# Tourism-Related Needs in the Context of Seniors’ Living and Social Conditions

**DOI:** 10.3390/ijerph192215325

**Published:** 2022-11-19

**Authors:** Klaudia Przybysz, Agnieszka Stanimir

**Affiliations:** Department of Econometrics and Operational Research, Wroclaw University of Economics and Business, 53-345 Wroclaw, Poland

**Keywords:** senior tourist, tourist activity, accessible tourism, withdrawal and exclusion from tourism, active ageing, importance analysis, correspondence analysis

## Abstract

Significant changes are taking place in the structure of tourism participants. Due to the ageing of societies, the tourism sector has to respond to the increasing tourist activity of seniors. The main aim of our research was the recognition of the needs of senior tourists from selected regions of Poland, considering their health and financial situation as well as their physical activity. The study shows how to combine the knowledge of assumptions of active ageing with the actual views of senior tourists on tourism and active leisure. An additional objective was to determine the reasons why seniors gave up tourism and to compare the reasons why seniors from selected regions of Poland and seniors from other European countries did not participate in tourism. Based on Eurostat data, we identify the most common reasons for people not participating in tourism who are over 65 years of age. In 2020, we surveyed seniors. The respondents for the sample were selected as 65 years and older. In order to compare countries due to exclusion and non-participation of seniors in tourism, the results classification was used. To analyse the touristic behaviours of Polish seniors, we used correspondence analysis. As indicated by analysing the reasons for the non-participation of Europeans aged 65 and over in tourism, in most countries, financial and health reasons are ranked first or second in 2016 and 2019. In a survey of Polish seniors, except for the financial reasons responsible for non-participation in tourism, an additional obstacle was the language barrier in foreign tourism. The analysis of physical and tourist activity showed that non-participation in tourism is associated with low physical activity. Women reported that they were satisfied with their financial independence and most often used the opportunity of short-term tourism. The people who are fully or largely involved in organising their trips also willingly change their locations during their next travels.

## 1. Introduction

In today’s societies, the importance of elderly people in different areas of life is changing significantly. Structural changes in the population resulting from the growing number of older people affect not only age-related social policy (mainly health), but also social, cultural, tourist and many other areas. Indirectly, this involves changes in the necessary infrastructure, social and health services, the offer addressed to seniors, as well as changes on the labour market. Therefore, it is crucial to undertake exploratory, preparatory and implementation activities in the sphere of social and economic functioning of the elderly [1,2]. They should not be limited to regional or national actions as the changes are visible in all corners of the world. The problem of ageing societies is global.

In our article, we focused on identifying seniors’ tourism activity. Their interest in tourism determines the attractiveness of the travel market on a global basis [3,4]. The main aim of our research was the recognition of the needs of senior tourists from selected regions of Poland, considering their health and financial situation as well as their physical activity. An additional objective was to determine the reasons why seniors gave up tourism and to compare the reasons why seniors from selected regions of Poland and seniors from other European countries did not participate in tourism. The following research questions were posed:RQ1:Does the activity of seniors (both physical and tourist) depend on their health and financial situation?RQ2:Is the distinction of seniors as a group of people who are at least 65-year-olds sufficient to determine tourist preferences?RQ3:What are the reasons for not participating in tourism?RQ4:Are seniors homogeneous in terms of not participating in tourism?RQ5:Is it possible to define patterns of tourist behaviour of seniors depending on their stay and destination preferences, considering demographic characteristics other than just age?

Our research was conducted in a twin-track approach. We will present the results of a questionnaire survey conducted among Poles aged 65 and over on their perception of tourism and recreation as well as their participation in these activities. We will also present the results of a comparative study of seniors from the European Union countries on exclusion from tourism and ways of using tourism in terms of demographic differences.

The Active Ageing report [5] highlights the importance of the requirement to simultaneously meet the needs of senior citizens, particularly physiological safety, belonging and respect. Health and welfare, behavioural and individual factors, external environment, social and economic factors should all be recognised as determinants of active ageing. In the presented study, we assumed that tourism is conditioned by its participants’ standpoint as well as by its ability to meet all their needs, spiritual changes, and social development, all related to active ageing. Olsson and Schuller [6] pointed to the individual perception of the connection between globalisation and the standard of living. They also pointed out that the individual’s perception of well-being is related to a specific moment in life. For this reason, it can be indicated that individual subjective assessments often result not only from the valuation of the actual standard of living, but also from comparisons that citizens make with regard to living standards in other regions of the country or in other countries [7]. This approach shows that the direct opportunity to assess the standard of living of other people during a trip creates a subjective rating of one’s own well-being, and consequently, a desire to improve its level in the future. A temporary subjective perception of widely understood well-being in the sense of participation and inclusion plays an important role here [8].

In studies related to active ageing, the authors repeatedly emphasise the growing importance of this phenomenon. Adding to this the significance of individual activity understood as senior tourism, this phenomenon becomes multi-faceted. Two areas of influence by senior tourism can be identified: individual and socio-economic. In the first area, factors directly influencing the elderly as a social actor should be identified. In the other one, factors influencing the functioning of the various areas of the economy should be identified [2,3,9,10,11,12]. These are, generally, the health system and economic growth, the labour market, and the growth of the tourism market. These issues are synergistic (they are in synergy and function indissoluble); the more tourism is considered, the more important the industry becomes [2,13,14].

The article consists of six parts including the introduction, conclusions and discussion, limitations, and future suggestions. In the Introduction, the research goal and questions are formulated. The research background and literature review are included in Chapter 2 where the authors focus on presenting active ageing, quality of life, and seniors’ age determination exclusion. The Material and Methods chapter presents the scope of our research of Polish seniors and Eurostat data on seniors’ social exclusion in the field of tourism. In Chapter 4, we present the results of our twin-track analysis: tourist exclusion and non-participation of seniors and touristic behaviours of Polish seniors. In Chapter 5 we compare the results of our study with the achievements of other researchers. Finally, we present the limitations of the research.

## 2. Research Background and Literature Review

The growing tourist activity of seniors and a significant heterogeneity of this social group require constructing new trends in the tourism industry. The tourist expectations of seniors can provide a solid basis for initiating further economic and social development. It is necessary to create a specific tourist offer for this target customer group, which would be specially adapted to the elderly population, recognising both their socio-demographic characteristics and their future expectations [3]. Therefore, it is important that the available tourism products and destinations are adapted to the growing needs of current and future senior tourists, allowing for their disabilities and thus making it more accessible and elderly consumer friendly by addressing their specific needs [1,3,12,13,15,16,17].

### 2.1. Active Ageing and Quality of Life

The Active Ageing report [5] introduces three pillars of active ageing, i.e., health, safety, and belonging to society. Observing active ageing in the proposed areas requires understanding of the quality of life as a concept of well-being and positive functioning of individuals in the society [18]. Ageing as observed today is a challenge for the future [10]. To implement the assumptions of active ageing, one should strive for ageing to be perceived as a regular stage in human development, which should not be feared.

The people currently considered to be seniors have completely different opportunities than those who reached the same age several years ago [10,12]. This is because the financial, medical, and mental opportunities of active ageing are changing, along with its social perception. It follows that the current behaviour of seniors will be of great importance for their future lifestyle [4]. Their future well-being and thus the level of satisfaction will depend on how well they look after themselves now [10]. In addition, the quality of life is linked to the possibility of continuing current active life and maintaining positive social relations [19]. Reduced activity and an increased level of peace and quiet provided during a relaxing vacation improve mental and physical well-being [3,4,15]. In addition, travelling introduces changes in everyday life and raises individual expectations as to the improvement of their life quality [3,15]. For this reason, people participating in tourism seek to improve their economic and health situation [15]. Deciding to participate in tourism results from one’s ability to subjectively assess one’s quality of life. This assessment reflects the state of life satisfaction. It is based on individual opinions on various aspects and includes both one’s own household and the impact of other people’s opinions on the choices made by the individual [7].

Active ageing, well-being, and quality of life, along with tourism related activities, necessitate the observation of functional balance. They all facilitate a balanced life which also encompasses mental and spiritual development [20]. Thus, it can be determined that to motivate seniors to adapt active ageing, the tourism of seniors should be defined as the realisation of all kinds of private purpose trips. It should also include a set of conscious decisions undertaken precisely in order to maintain functional balance and well-being. Its effect is to change seniors’ intellectual and emotional perception of themselves in the surrounding society. This would involve higher self-esteem, more respect, admiration, and self-improvement [1]. Seniors’ tourism, understood in this way as an aspect of life quality, meets the definition used in Eurostat [21], where one of the areas of its measurement is leisure, social interaction, and general life experiences.

Encouraging seniors to participate in tourist trips may not start at the time of their retirement or when they reach the age recognized as senior in a given country. Tourism’s position in meeting the needs of seniors depends on the way in which social life is realised over the age of 65 as a result of being active in an active age. It is extremely difficult to change the existing, one might say acquired, habits of the elderly. People who were homebodies at their active age (which resulted from their conscious choice, not necessity or duty), will not become travelling seniors.

When looking at tourism of seniors, one should be aware of the requirements they place on their travel conditions. With age their needs related to the destination increase. It must provide adequate accommodation, social facilities, and a safety guarantee. These factors significantly emphasise the imagined journey. Seniors, taking the touristic needs and destinations into account, define in detail the necessity to travel and identify travel as a product. Then, they make a real journey, and after their return, their tourist activity is focused on the already remembered journey.

### 2.2. Seniors’ Age Determination

To combine active ageing with participation in tourism, it is necessary to consider the age of a senior tourist. First of all, it is necessary to refer to the pan-European definitions. By the European Commission [22], older people are indicated as indirect beneficiaries of the elderly services economy (care, health, and other age-friendly environment services in long-term) and innovations for active and healthy ageing [23]. In the European Platform against Poverty (Flagship Initiative of European 2020 Strategy) elders appear as a part of the society at risk of poverty [24] and as a social group against which protective measures should be taken [22]. Many authors justify the correct definition of age groups in different ways. According to Hossain, Bailey, and Lubulwa (cited in [2]), elderly people who are an important tourist segment are 50–75 years old. Alén et al. [15], after studying the literature, presented the following groups: elderly tourist—over 50 (or 65) years old (in some studies, in other studies only until the age of 74 years), senior tourist over 55 (or 60) years old. They also indicated that seniors are aged 55 years or older, where the following can be distinguished: younger elderly subgroups (55 to 64 years old) and the older elderly (65 years old or more). The ages used to identify senior citizen tourists vary depending on the country and the socioeconomic conditions under consideration. Przybysz et al. [25] found that an officially older adult is a person over 60 years of age, but in the Central Statistical Office of Poland studies, one can find a reference to Eurostat and OECD, in which the term elderly is used for people over 65.

It is not sufficient to indicate age as a determinant of the group of senior tourists, because there are large differences in socioeconomic characteristics resulting from age, which implicates a strongly heterogeneous group. Regardless of the lower age limit for senior tourists, it is important to point out that current seniors are increasingly aware of the importance of active ageing, but also healthy ageing and a healthy lifestyle (including diet). The above-mentioned factors mean that the group of senior tourists is not homogeneous in terms of interests, financial and health opportunities, requirements, lifestyles, and levels of education [2,11,15]. However, it is a group with greater purchasing power than previous generations of seniors. They expect 4A’s: Attractions, Access, Amenities, and Ancillary Services [11]. At the same time, they are now more critical of the (tourist) industry due to their extensive experience in tourism acquired in their active age [15]. Therefore, to identify senior tourists except for the age [4], it is also necessary to consider their consumer behaviour, given their physical and mental condition [13].

### 2.3. Seniors’ Exclusion

An important aspect in the discussed area of senior tourism is a negative phenomenon of social exclusion in the field of tourism. Referring to the functional balance, it should be pointed out that in their pursuit of active life, seniors should be given support to recognise the age of 65 as an onset of active time not just a moment of their retirement. Muras and Ivanov [26] indicate, referring to the definition of the European Commission, that social exclusion is generally defined in terms of non-participation or inability to participate in important aspects of collective life: social, economic, political, and cultural, and as non-participation in normal activities characteristic of a given society. According to this definition, it should be determined whether we are dealing with inequality or exclusion in tourism.

In addition, there is another important concept to be considered, especially in the field of seniors’ tourism, which is social withdrawal. It is a result of a steady, gradual withdrawal or exclusion of older people from the social roles they performed during their active age. These actions are inevitable. However, they do not have a meaningful negative overtone when new roles and social relationships appear instead of the existing ones, which is in line with the aforementioned adaptive attitude in ageing [1,10]. Considering the broadly understood health, mental, and social reasons, the inability to participate in tourism can be understood as the exclusion of seniors from tourism. On the other hand, financial considerations will direct non-participation in tourism towards inequalities in tourism.

Nimrod and Rotem [27] indicated that travelling may be a challenge for seniors in terms of planning, solving unexpected problems, and coping with new situations. Overcoming this challenge can benefit the senior tourists in contact with their environment. The experiences gained during travels change their perspective on their living conditions, e the way they view themselves, and the way they are viewed by others. They also strengthen their sense of independence and freedom. Alén et al. [16] indicated that travelling is a method of improving self-esteem. It is very important in the context of negative suggestions from the closest environment and family members [1,28]. Often, health problems affect the financial affluence of seniors, due to the necessity of bearing the costs of the treatments. Thus, undertaking a trip by a senior is met with the fears of the family and the environment about the possible deterioration of their health. When they participate in recreation and tourism they overcome (reasonably) their own limitations, which may bring additional benefits. Accordingly, it is clear that the concept of ageism has an essential meaning for analysing discrimination against seniors [25]. In creating his definition of ageism in 1969, Butler [29] intended to present a number of factors focusing on prejudices against the elderly. Thus, the animosity of the environment towards seniors and tourism results from negative stereotypes and, as indicated by Vauclair et al. [30], the concept of ageism covers all of these aspects. Therefore, while encouraging tourism, it is necessary to counteract age-related stereotypes and stigmatization regarded as soft discrimination [25].

When analysing the quality of life of seniors through the prism of their participation in tourism, attention should be paid to why they engage in such an activity. The reasons have appeared most frequently in previous studies. The most common motives for travelling by seniors in the research described in the literature are: visiting family or friends, social contacts and participation in social life, searching for new things, searching for knowledge, and escaping from everyday life [3,13,15,16,31,32,33]. An interesting determinant of travelling is taking care of the continuity of life [32].

Concluding on travel determinants and exclusion in tourism, it is worth relating the tourism and recreation needs to Maslow’s hierarchy of needs, which classifies them as higher-order needs which are related to self-actualization, love and belongingness, and physiological needs [33]. However, it should be remembered that for seniors, as there are changes in their social role, personality, life experience, and relatively stable and certain income, the need for hierarchical fulfilment of needs disappears and the fulfilment of physiological needs, security, belonging, and respect must be met simultaneously [1].

## 3. Materials and Methods

In our study, we considered a significant indication that it is not only seniors in Poland who do not participate in tourism, but also seniors from other parts of Europe. To indicate the fundamental reasons for the resignation of European seniors from tourism, we used Eurostat data from 2016 and 2019. We took the analysis of the non-participation of seniors in tourism as the background to the main study. The data available in Eurostat is aggregates and not individual data. Data on non-participation in tourism has been collected so far for 2013, 2016, and 2019. The 2019 data was re-leased in July 2021, exposing a significant delay. Eurostat data is representative (as shown by the metadata file). The surveyed population is people aged 15 and over. The age range is predetermined. Based on this data, it is possible to identify the most common reasons for people who do not participate in tourism and who are over 65 years of age. With regard to the above-mentioned groups of seniors, it was also possible to examine the reasons for non-participation in tourism among those aged 55–64. However, we decided that these people, compared to the elderly, are not so much at risk of being excluded from tourism because more often they remain professionally active and their health is better. In 2016, according to Eurostat data, 48 million people aged 65+ did not participate in tourism, and in 2019,44 million people of this age did not participate in tourism. They constituted 50% and 43% of people in this age group in the European Union, respectively. In our analysis, we did not take into account the data from 2013 due to the large time gap between 2013 and the survey we conducted.

Eurostat data on non-participation in tourism is complete data in the area established by Eurostat. A drawback of the data available in Eurostat is the inability to check the situation of non-participation in tourism by seniors, broken down into different demographic groups depending on education, place of residence, etc. The second source of data that we used allowed us to identify and indicate the reasons for non-participation in tourism, depending on various demographic characteristics.

In 2020, just before the outbreak of the COVID-19 pandemic, we conducted a survey among seniors. The respondents were selected in a sample of people aged 65 years and older. The survey was conducted in three regions of Silesia in Poland (NUTS2): Dolnośląskie (PL51), Opolskie (PL52), and Śląskie (PL22) (Figure 1). It is the southwest part of Poland which borders Germany and the Czech Republic. There are two airports in this region serving flights to most European countries and airports in Poland. There is also a very dense railway network. It is possible to connect with the countries of the south and west of Europe and with many attractive tourist destinations in Poland. In addition, the A4 and A1 motorways run through these three regions.

A snowball technique was applied when choosing the next interviewee. The choice of this form of research was based on several crucial aspects. Older people are reluctant to fill in questionnaires and participate in studies. The number of questions and the complexity of the questionnaire may also contribute to their reluctance towards a survey. The snowball technique reduces the respondents’ fear of the survey. Sixty-eight respondents took part in the survey, one of whom refused to give answers. The structure of the respondents is presented in Table 1. The number of men participating in the survey reflects the changes in the demographic structure with the ageing of the population.

In order to verify the research questions and realise the postulated research problem, correspondence analysis and results classification were used. Correspondence analysis (CA) has been widely described in the literature, mostly in many publications by Greenacre [34,35], as well as those by Backhaus et al. [36], Blasius [37], and Lebart et al. [38]. This article does not present the algorithm of CA, but merely refers to the special construction of data tables, indicators, and assumptions regarding the correctness of the conclusions reached.

The results classification method allows grouping the results achieved for the survey objects. Thus, descriptive statistics are calculated for the aggregate index determined during the study. Then, based on the Nowak [39] approach, the data has been classified into four groups based on the mean and standard deviation (*st. dev*):-1st most important: higher or equal than mean+st.dev;-2nd very important: ⟨mean;mean+st.dev);-3rd important: ⟨mean−st.dev;mean);-4th least important: lower than mean−st.dev.


The first group includes countries where a given cause is indicated as the most important for the survey phenomenon or with the highest values of the aggregate index, and the fourth group is the least important or with the lowest values of the aggregate index.

Using CA, it is possible to examine the relations between the categories of at least two non-metric variables. The result of this method is an indication of the groups of coexistent categories and their graphic presentation. In the basic scope, CA analyses the relations between the categories of variables included in the contingency table. In order to perform graphical analysis, a singular values decomposition (SVD) of the correspondence matrix is used. On this basis, the coordinates of rows’ and columns’ categories of the contingency table are determined. To verify the results of the CA, the eigenvalues, which are squares of the singular values, are used. The sum of the eigenvalues is the total inertia. One-dimensional space is related to the first eigenvalue, two-dimensional space with the two highest eigenvalues. Thus, it is appropriate to check what percentage of total inertia are eigenvalues in a descending order.

CA for small samples is described in detail in [37]. To carry out this procedure, it is necessary to build an indicator matrix. This matrix, as Blasius and Greenacre [40] (p. 27) wrote, “is a respondents-by-categories table with as many rows as respondents … and as many columns as response categories”. The only elements of this matrix are zeros and ones. Ones identify the respondent’s choice, and zeros appear in other places. The indicator matrix is called the matrix of dummy variables, and in the case of many variables, it is called the superindicator matrix. The CA of such a matrix is carried out according to the classical approach, i.e., as for the contingency table, using the SVD. The full space for analysing the relationships of the variable categories is *K*-dimensional:(1)K=min(r−1;c−1),
where *r* is the number of respondents, *c* is the number of all categories of analysed variables.

Additionally, in CA it is possible to use passive variables which do not take part in determining the solution space; only active points have the influence on the geometric orientation of the axis. Passive variables support the interpretation of the configuration of active elements [40]. Passive variables (additional points) illustrate the information that is not presented in the analysed interactions of the categories of active variables. The coordinates of passive variables are determined on the basis of singular values and singular vectors determined during the analysis of active variables. The inclusion of passive points in the analysis occurs only after performing all calculations for the variables whose relations are examined (active variables).

## 4. Results

### 4.1. Tourist Exclusion and Non-Participation of Seniors

Eurostat data shows what percentage of the population in a given age does not participate in tourism because of financial reasons, lack of interest, lack of time due to family commitments, lack of time due to work or study commitments, health reasons, safety reasons, and others. The data we compared are from 2016 and 2019. For each year, data have been classified into four groups. The first group includes countries where a given cause is indicated as the most important, and the fourth group is the least important. In some countries, these data are not available (Sweden and the United Kingdom in 2019) or indicated as confidential (other countries in 2016 and 2019). Results are presented in Table 2.

Among the reasons for not participating in tourism analysed in Table 2, there are those that are most often indicated in the literature: financial concerns and health. Financial reasons were most often quoted by Bulgarians, Greeks, and Portuguese in 2016 and 2019. In Poland and France, this reason was also the most important despite the fact that in 2016 these countries were in the second and third groups, respectively. In France, an increase in the importance of no interest in travelling (from the 3rd group in 2016), no time due to family commitments (from the 2nd group), and safety reasons (from the 3rd group) as reasons for not participating in tourism were also recorded in 2019 compared to 2016. For the Portuguese, apart from financial reasons, no interest and no time due to family commitments were the most significant reasons for not participating in tourism in both 2016 and 2019. These two reasons were also the most frequently chosen by Slovaks in 2019. In the case of no time due to work or study commitments in 2016, Hungarians, as well as Lithuanians in 2019, most frequently chose this as the most important factor of their non-participation. The latter reason for not participating in tourism was chosen the least frequently by seniors from the analysed countries in both years. The analysis of the reasons for not participating in tourism shows that the decisions of Poles allowed them to be classified in the second group in 2016, except for a lack of time, whether for familial or professional reasons. As we have already mentioned, in 2019, financial reasons were most often chosen by Poles, while in the assessment of other factors, they fell into the second group (no interest, no time due to work or study commitments, health reasons), and the third group (no time due to family commitments, safety reasons). When analysing the reasons, it is not possible to unequivocally identify the country where inhabitants give many reasons for not participating in tourism, and these opinions remain constant over time. It is also impossible to define groups of countries assessing causes similarly. It follows that seniors in individual countries individually define their attitude to tourism, and their economic and health situation changes over time. Determining the level of the quality of life of a particular older person requires looking at their life and surroundings from their individual perspective. Subjectivism in assessing the quality of life is universally important but it is of particular importance in the case of seniors as there is a re-evaluation and change of priorities of implemented activities and needs, e.g., issues regarding retirement income are not so urgent because there is the certainty of a guaranteed income.

Summing up the considerations on the exclusion of seniors in tourism, it is worth pointing out that in both years, the most frequently mentioned reasons were financial and health reasons, and lack of interest was the reason for touristic withdrawal. Therefore, seniors are a group at risk of being dissatisfied with life as a result of not participating in leisure activities.

### 4.2. Touristic Behaviours of Polish Seniors

The tourist behaviours of Polish seniors were examined based on the questionnaire presented in Appendix A.

A structured questionnaire (Appendix A) was developed to identify the key characteristics of seniors’ perceptions and involvement in tourism and their physical activity. The questionnaire consisted of three main sections that examined: physical activity and the willingness of seniors to travel (Q1–Q6), then the reasons for not participating in tourism (Q7–Q9), and finally the behaviours of senior tourists (Q10–Q13). The questionnaire ends with socio-demographic questions. This study adopts mainly correspondence analysis to investigate the characteristics of seniors as travellers.

For all the questions where respondents did not express their opinion in order to ensure compliance with the small sample CA requirements, a category of zero (e.g., Q3.0) was introduced.

In the first analysis are three variables describing tourist and physical activity (Q1, Q2, Q5) and five demographic variables (gender, age, marital status, level of education, place of residence) are active variables in the analysis. Additionally, we have three passive variables (Q3, Q4, Q6). The real relationships between the categories of the mentioned variables can be described in the R^25^—25-dimmensional space (only 16 eigenvalues are higher than zero). If the variables were completely independent, each eigenvalue should be equal to $\frac{1}{Q}=\frac{1}{10}$ (where *Q* is the number of active variables). In our analysis, the first two eigenvalues are higher than 0.1. Using the correspondence analysis, the dimensions of the relations of variables categories from R^25^ to R^2^ were reduced. The first two eigenvalues of 0.24 and 0.2 account for 14.7 and 12.6 per cent of the total inertia (Figure 2). While the first axis can be considered as defining physically and touristically active seniors (activity increases from the left to the right), the second axis appears for younger seniors on lower side, and for the oldest seniors on upper side.

The answers of seniors who use tourist offers dedicated to them (Q1.1) coincide with the answers of those who rate their health very well (Q3.1) (Figure 1). Respondents who spend free time very actively (Q5.1) rate their health well (Q3.2) or did not indicate that they do not use tourist offers for seniors in their place of residence (Q2.1). Women indicated that they make short trips as a form of recreation once a week (Q6.2), but also more often (Q6.1) and spend free time actively (Q5.2). Respondents from big cities make short trips and go for a walk more often than once a week (Q6.1). Seniors very satisfied with their financial independence (Q4.1) have higher education (Edu1). They did not indicate that they do not use tourist offers (Q1.3-). The location of points describing the youngest seniors living in medium sized towns pointed out that they rarely spend free time actively (Q5.4). Seniors who are rather satisfied with their financial independence (Q4.2) go for walks and take part in short trips only once a month (Q6.4). Seniors who define their health status as bad (Q3.4) and never use tourist offers for seniors (Q1.1-) also never benefit from touristic offers in their city (Q2.2) and sometimes they define their health status as slightly better (Q3.3). This is characteristic of seniors in relationships (not single). The group of seniors with the education level lower than high (Edu2) rate their financial independence as bad (Q4.4). Middle aged seniors (Age2) spend their free time actively on average (Q5.3), and their financial independence is neither good nor bad (Q4.3). Another group we want to describe are seniors who never use generally available tourist offers (Q1.2-). They rate their health and financial independence very bad (Q4.5) and they are inactive (Q5.5) or rate their health status as very bad (Q3.5). Finally, we want to indicate the location of the point Q1.3+, which illustrates people who do not use tourist offers. It is very far away from other active and passive points, which means that these people who do not participate in tourism (for various reasons) are very different from the people who want to participate in tourism.

Among the respondents, 38 people indicated that they do not go on domestic holidays or trips, and 35 people do not go abroad for such a reason. Not participating in one of the indicated forms of tourism did not exclude participation in another. Seniors not participating in domestic or foreign tourism gave reasons for their choices (Table 3).

Table 3 shows what proportion of each demographic group are people covered by the given reason for not participating in tourism (on basis of Q7, Q8, Q9). Forty-three percent of men, a significant part of people aged 70–74, indicated resignation from foreign tourist trips due to the language barrier and financial situation. This reason was more often given by single people with a lower level of education and living in small towns. In the case of resignation from domestic trips due to financial issues, the differences between men and women were not so great. Due to the financial situation, more people who are not single and live in big cities and do not have higher education have resigned from these types of trips. It is worth noting that a similar percentage of people aged 65–69 and 70–74 indicated financial issues as the reason for giving up a domestic trip, with a very low percentage of the oldest people giving the same reason.

Background information about the demographics of the respondents as well as their travel preferences is presented in Figure 3.

Figure 3 shows the results of CA of variables Q10–Q13 and demographic variables, accounting for 20.95% of inertia. As the location of the Age1 point is in the centre, it is not possible to indicate any additional characteristics for this group of seniors. We will start the description of the other relations between the preferences of seniors with the upper side of the second axis clockwise.

Seniors who never use holiday stays offered by travel agencies, even with their own transport (Q10B.5, Q10C.5), or never use public transport to get to the places of their choice (Q10D.5), stay on domestic vacations for the longest (Q11A.3) (Figure 3). Seniors who never organise their travel on their own (Q10A.5) rarely change places of stay during domestic trips (Q13A.3). The next three singled out groups are people who are not interested in travelling or did not indicate their preferences in this matter. In the first group are men with lower education who during domestic holidays rarely change destination (Q13B.3) and did not provide their opinion on destinations of domestic excursions (Q13A.0), long and short foreign trips (Q13C.0, Q13D.0), domestic and foreign length of stays, or frequency of trips (Q11A/B.0, Q12A/B.0). Non single people aged 70–74, from small and medium-sized towns, did not give the information about their preferences of destinations during domestic holidays (Q13B.0). Seniors who during domestic holidays never change destination (Q13B.4) did not inform about how they organise their trips (Q10A.0–Q10D.0). Those who rarely use the offer of a travel agency and organise their travel on their own (Q10C.3), go on domestic holidays for seven days (Q11A.1) once a year (Q12A.1) and always change places of stay during domestic holidays (Q13A.1). Moving to the third quadrant of the chart, we get groups with more diversified travel preferences. For example, people who often use the offers of transport providers and organise their stay on their own (Q10D.2) also often use comprehensive services of travel agencies (Q10B.2) and rarely organise the entire trip on their own (Q10A.3). The next group includes people who always use comprehensive services of travel agencies (Q10B1) or rarely use such offers when they have to get to their place of stay on their own (Q10D3), their trips abroad last over two weeks (Q11B.3), and they always choose new places to travel abroad (Q13C1). Among seniors who rarely use comprehensive services of travel agencies (Q10B.3) (and if they do so, they always or often use only the stay offer (Q10C.1, Q10C.2) and travel with a transport provider (Q10D.1)), their domestic trips last 8–14 days (Q11B.2), they travel abroad once a year (Q12B.1), and always choose new places for holidays abroad (Q13D.1). Highly educated seniors from large cities organise trips on their own (Q10A.1), go for 8–14 days on domestic holidays (Q11A.2), and for 7 days abroad (Q11B1), but twice a year (Q12B.2). People who often change destinations, both domestic and foreign during all short and long trips (Q13A.2, Q13B.2, Q13C.2, Q13D.2), often organise the entire trip on their own (Q10A.2), and often during the year, they go on domestic holidays (Q12A.3). They rarely choose new places for holidays abroad (Q13D.3). The youngest, single women indicated that when they go abroad, the trips last seven days.

## 5. Conclusions

This article focuses both on the perception of tourism and physical and tourist activity of the elderly, as they both reflect the assumptions of active ageing. Based on the literature review, it was found that the most common reasons for non-participation and withdrawal of seniors from tourism are financial and health reasons. We were looking for determinants of a physical and tourist activity. Moreover, we wanted to indicate what additional characteristics can be specified for senior tourists in relation to their specific decisions about the number of trips, the length of stay, the destination, and the form of organising their trips. This research provides some implications for other studies, but at the same time, it presents a picture of the society that can benefit stakeholders in the tourism industry.

In the literature, some reasons for travelling are given by seniors. They focus on the fulfilment of needs and motives for action such as visiting family or friends, social contacts, and participation in social life, which are all closely related. When choosing a travel destination, seniors pay attention to climate and weather, especially a pleasant air temperature [41]. Grześkowiak et al. [1] and Liew et al. [4] indicate the importance of safety, hygiene and cleanliness of the accommodation. A certain attempt at classifying destination choices is grouping them into universal and specific. The universal or key reasons are: unique natural [11,13], scenic resources [11,13], comfortable climate [11], safety of tourism attractions [11], relaxation, well-being, socialisation and self-esteem [13]. Specific factors include barrier-free: public transportation, accommodation, and facilities along travel routes [4,11,33]. Factors such as self-development and relaxation [13] as well as physical activity [33] are evidence of the changing perception of tourism by seniors towards more active pursuits with a focus on health and fitness.

The types of tourism chosen by seniors also influence the motives for tourism choices. Meyer [42] indicates that the most popular among people aged 65 and over are leisure, cognitive, health-related (spa, fitness, and wellness), religious (religious and cognitive, cognitive and pilgrimage) and ethnic (sentimental) tourism. McCabe and Qiao [43] distinguish social tourism popular among seniors struggling with disabilities and age-related limitations. This type of tourism supports the pursuit of happiness, satisfaction, health, and social inclusion. Tourism of seniors is social tourism because seniors are a group that may be excluded from participation in regular tourism due to the limitation or difficulty of access [13,44].

Research conducted among Polish seniors indicates that they do not travel for financial reasons [44,45,46,47] or for health reasons [45,46,47]. This conclusion is in line with the results of our study. However, the survey we conducted among selected seniors showed yet another reason: a language barrier. The authors of many studies indicate that seniors choose domestic stays, which they organise themselves or with the help of travel agencies [44,45,46,47]. Seniors in other countries do the same, making additional use of the Internet [48]. The purpose of trips by Polish seniors is leisure and family reunions [49,50]. Leisure tourism also combines cognitive values [45,46,47,49,50]. The seniors participating in our study gave us similar feedback. However, we also indicated differences between seniors resulting from socio-demographic characteristics. Among other things, an important observation was that active people make active tourists.

As indicated by analysing the reasons for non-participation of the Europeans aged 65 and over in tourism, in 2019 there was a change in the reasons given by seniors in the EU member states compared to 2016. Only in Portugal, both in 2016 and 2019, seniors gave the same three reasons (financial reasons, no interest, no time due to family commitments), and the same two in Germany (no time due to family commitments, safety reasons) and Slovakia (no interest, health reasons). In other countries, the importance of the above-mentioned reasons changed. This is probably due to the generation-shift described in the article. Both the financial opportunities and the form of activity of seniors who entered the 65+ group in 2019 changed compared to those who were in this group in 2016. It can also be noted that in most countries, financial and health reasons are ranked first or second in 2016 and 2019. In our survey, except for the financial reasons responsible for non-participation of Polish seniors in domestic and foreign tourism, an additional obstacle appeared. The language barrier was the reason for non-participation in foreign tourism for 43 per cent of men, 47 per cent of people aged 70–74, 48 per cent of people without higher education and 38 per cent of people living in small towns. For these people, compared to other demographic groups, financial reasons for not participating in tourism were also indicated. Financial issues were often given by single people and people living in small towns as the reason for not participating in foreign tourism, while for domestic travel that reason was given by non-single and people from big cities.

The analysis of physical and tourist activity showed that non-participation in tourism is associated with low physical activity. Polish seniors who indicated that they use tourist offers generally available to all declared at the same time that they are actively spending their free time and that they are satisfied with their health condition. The respondents who indicated health problems did not use tourist offers generally available and they are less active. People with higher education were more likely to participate in a tourist offer in their place of living and they were very satisfied with their financial independence. A very interesting characteristic was observed for women, as they reported that they were satisfied with their financial independence and that they most often used the opportunity of short-term tourism also organised in their place of residence.

The analysis of tourist preferences has shown that people who are fully or largely involved in organising their trips are also willingly change their places of stay during subsequent trips, the number of their domestic and foreign trips is greater, but the duration of their stay is maximum of 14 days. Such tourist decisions are characteristic of people from big cities and with higher education. In order to know the exact conditions of such choices, it would be necessary to get to know the financial situation of seniors or the tourist habits they had when they were professionally active. Women are more active in tourism than men.

Tourism is not only an activity aimed at discovering new places or leisure. It also includes sociological, economic, and cultural elements. The incentives received during the journey motivate the body to further strenuous activity, thus increasing well-being and psychophysical comfort. However, it should be remembered that in the occurrence of significant difficulties, a senior tourist may become discouraged; “Older people are the experts of their own lives” [51] (p. 70). Therefore, one should carefully recognize their tourist expectations. However, to avoid generalisations, every stage of senior age and its corresponding needs should be considered separately.

## 6. Limitations and Future Suggestions

The contribution of the presented study to the discussion on seniors’ tourism preferences and exclusion in tourism is significant. However, there remain aspects of senior tourism that have not been explored and remain important. Thanks to the data collection method (snowball) we believe that the respondents of our questionnaire have exposed their actual preferences in tourism destinations, travel behaviours, and physical activity. Using our questionnaire with a larger group of respondents from the population of senior tourists will allow for an in-depth analysis. In such a situation, the research with the use of more personal factors related to senior tourists’ behaviours will be possible.

Both in the literature research as well as in our study of Polish and European seniors, the importance of health aspects can be noted. They are responsible for the exclusion or inclusion of seniors in tourism. However, the health factor as a reason for the reluctance to be active in tourism is, looking at from another angle, a factor in the success of changes and developments in medicine and the increased awareness of health-promoting behaviour. Perhaps, therefore, this factor is in many cases an imaginary factor or a stereotype carried by generations. Seniors’ mobility is a key to a healthy and active lifestyle [52]. Therefore, health and tourism are tightly combined.

Current active generations should be monitored so that a quick response is possible in the future. Their purchasing patterns and preferences with regard to tourist services should be recognized, as they will have similar expectations in the future [3].

We believe that if you want to develop the tourism industry, you should pay attention to the availability of the tourist offer for seniors and modify it to their needs. Another aspect of developing opportunities for senior citizens to participate in tourism abroad is to encourage them to learn foreign languages. At the same time, it should be remembered that the offer directed to seniors should be as attractive as the one directed to young people. No shortcomings of age or physical condition should be emphasised. Adapting the tourist offer to the requirements of seniors has been discussed many times in the literature [45,46,47,53]. The tourism industry cannot determine the needs of seniors by itself; it can only do so with significant cooperation with seniors and constant monitoring of their needs and expectations.

Very important aspects related to tourism are tourism in relation to the sustainable development and behaviours that affect many areas [54,55,56,57].

## Figures and Tables

**Figure 1 ijerph-19-15325-f001:**
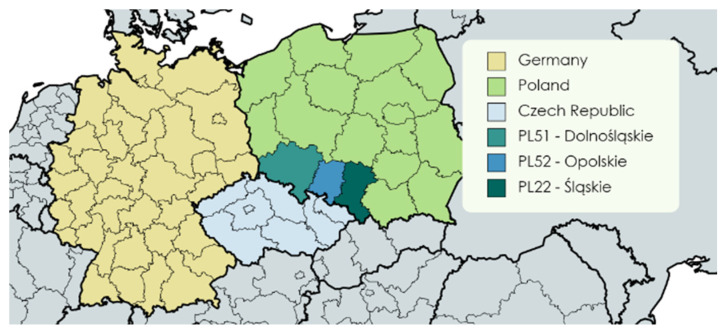
The surveyed regions of Poland (NUTS 2): Dolnośląskie (PL51), Opolskie (PL52), and Śląskie (PL22). Source: Own elaboration.

**Figure 2 ijerph-19-15325-f002:**
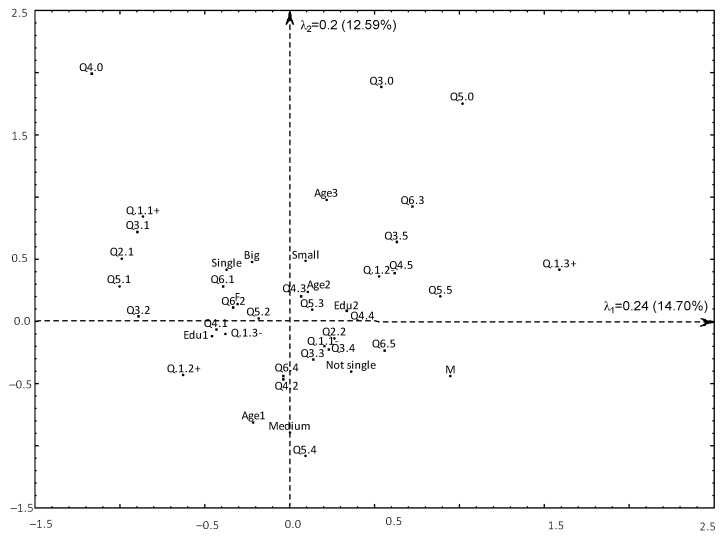
Touristic and physical activity of seniors. Source: Own computation using Statistica13.

**Figure 3 ijerph-19-15325-f003:**
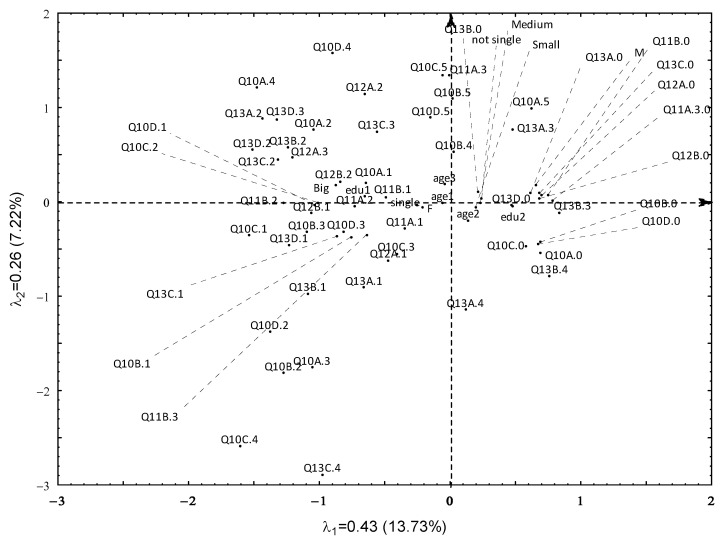
Seniors travel preferences. Source: Own computation using Statistica13.

**Table 1 ijerph-19-15325-t001:** Respondents’ profile.

Demographic Characteristics	Frequency	Per Cent of Total
Gender		
Female	43	75
Male	14	25
Age (years)		
65–69	25	44
70–74	15	26
75 and above	17	30
Marital status		
Single	29	51
No single	28	49
Level of education		
Higher education	24	42
Other	33	58
Place of residence		
Small town	26	46
Medium town	20	35
Big city	11	19

Source: Own study.

**Table 2 ijerph-19-15325-t002:** Importance of reasons for the non-participation of Europeans aged 65 and over in tourism in 2016 (16) and 2019 (19).

Group/ Year		Financial Reasons	No Interest	No Time Due to Family Commitments	No Time Due to Work or Study Commitments	Health Reasons	Safety Reasons
1	16	BG, GR, PT	BE, PT, SK, SE	AT, BE, DE, PT, IT	HU	HR, CZ, SK	AT, DE
	19	BG, GR, PT, PL, FR	FR, AT, PT, SK	DE, FR, IT, PT, SK	LT, HU	HR, CZ, SK	AT, IE, FR, DE
2	16	BE, HR, CY, EE, LV, PL, RO, SI, SK, SE, HU, UK	AT, EE, FI, IE, ES, NL, DE, PL, IT	CY, FR, ES, UK	HR, LT, PT, SI, UK	AT, CY, DK, EE, GR, ES, IE, LV, DE, PL, PT, SI	IE, LT, PL, PT, SK, UK
	19	BE, HR, LV, LT RO, SK,	BE, DE, IE, GR, ES, IT, LU, NL, PL, FI	ES, LV	PL, PT,	AT, CY, DK, EE, GR, ES, IE, LV, DE, PL, PT, SI, MT, RO	GR, LT, PT
3	16	DE, IE, ES, FR, IT,	HR, CY, CZ, GR, FR, LU, MT, SI, UK	BG, GR, NL, IE, LT, LV, PL, RO, SI, SE, HU	BG, CY, DK, FR, GR, ES, NL, IE, LV, PL, RO, SE, IT	BE, BG, FI, NL, LU, MT, RO, SE, IT	BG, HR, DK, GR, NL, FR, LV, RO, SI, SE, HU
	19	DE, EE, IE, ES, IT, CY, LU, HU, NL, AT, SI	CZ, DK, EE, HR, LV, LT, MT, SI	BE, BG, CZ, IE, GR, CY, LT, HU, NL, PL, RO, FI	BE, BG, DK, IE, GR, FR, HR, IT, CY, NL, PR, SI, FI	BG, DK, FR, LU, NL	BE, BG, DK, ES, CY, LU, HU, NL, PL, RO, SI, FI
4	16	CZ, DK, MT, NL, AT	BG, DK, LT, LV, RO, HU	HR, DK	–	FR, LT, HU, UK	–
	19	CZ, DK, MT, FI	BG, CY, HU, RO	DK, SI	–	BE, IT, LT, HU, FI	–
– –	16	FI, LU	–	CZ, EE, FI, LU, MT, SK	AT, BE, CZ, EE, FI, LU, MT, DE, SK	–	BE, CY, CZ, EE, FI, ES, LU, MT, IT
	19	SE, UK	SE, UK	SE, UK, AT, HR, EE, LU, MT	SE, UK, CZ, DE, EE, ES, LV, LU, MT, AT, SK	SE, UK	CZ, EE, HR, IT, LV, MT, SK, SE, UK

ISO 3166 country codes. Source: Own study; Eurostat data (tour_dem_npage).

**Table 3 ijerph-19-15325-t003:** Reasons for resigning from tourism (%).

Trip	Reason	Gender		Age			Marital Status		Education		City		
		F	M	65–69	70–74	74+	S	NS	H	O	S	M	B
F	L	23	43	24	47	18	31	25	21	48	38	20	18
	Fin	35	64	36	60	35	45	39	33	70	62	35	9
D	Fin	21	29	24	27	12	17	29	13	43	0	15	91

F (Foreign); D (Domestic); LB (language barrier); Fin (financial issues); 74+ (over 74); S (single); NS (no single); H (higher), O (other); S (small); M (medium); B (big). Source: Own study.

## Data Availability

The data that support the findings of this study are available from the corresponding author.

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
