# Peer review of "Tourism-Related Needs in the Context of Seniors’ Living and Social Conditions"

_ijerph, 2022, doi:10.3390/ijerph192215325_

Round 1

Reviewer 1 Report

The article presents is very interesting because research results related to the recognition of the needs of tourists' 10 seniors concerning their health and financial situation as well as physical activity. The study shows how to combine knowledge about assumptions of active ageing with actual views of senior tourists on tourism and active recreation. The principal objectives about this paper is identify the most common reasons for people not participating in tourism who are over 65 years of age. The authors are congratulated for the clear and accurate exposition of the theories used. The abstract is correct and include all de items more important. Also, the theoretical used is rich in bibliographic sources. The methodology is accurate and is well explained. Finally, the bibliography is adequate, numerous and up-to-date. For all these reasons I think that this paper is adequate for publication in this journal.

Author Response

Dear Reviewers and Dear Editors,

Thank you very much for all, very important remarks. We are submitting the article in the track changes mode to make it easier to identify all the items that have been improved, moved, or added to the original version of the article. Since the reviewers indicated various areas that we should improve, we tried to create a coherent version containing as many valuable suggestions from the reviewers as possible. We are very grateful for such detailed comments.

Response to Reviewer 1 Comments

The article presents is very interesting because research results related to the recognition of the needs of tourists' 10 seniors concerning their health and financial situation as well as physical activity. The study shows how to combine knowledge about assumptions of active ageing with actual views of senior tourists on tourism and active recreation. The principal objectives about this paper is identify the most common reasons for people not participating in tourism who are over 65 years of age. The authors are congratulated for the clear and accurate exposition of the theories used. The abstract is correct and include all de items more important. Also, the theoretical used is rich in bibliographic sources. The methodology is accurate and is well explained. Finally, the bibliography is adequate, numerous and up-to-date. For all these reasons I think that this paper is adequate for publication in this journal.

Dear Reviewer. Thank you very much for your evaluation. We are glad that our article met with your approval. The analysis of the seniors' tourist activity is relevant to the physical and mental health of the elderly as well as to the improvement of their life quality. The results of the analysis may also benefit the tourism industry, by fostering its adequately targeted development. For this reason, we sincerely thank you for accepting our article for publication.

Reviewer 2 Report

Generally, due to two different data sources, I suggest that their characteristics should be
included in sub-section 4.1 (Eurostat) and section 4.2 (questionnaire survey).

In section 3, I suggest only a general comment that data sources, due to their diversity, will be
discussed in the following sub-sections of the article, respectively in sub-section 4.1 (data
obtained from the Eurostat database) and 4.2 (data collected on the basis of the Authors' own
questionnaire, including the data collection method, the sample size, the geographical scope,
etc.). It will surely be more understandable for the reader then.

Ad 3.

Please divide the section on the literature review as suggested earlier. I do not assume the
extensive expansion of this section with new elements added, but mainly the identification of
thematically coherent points. I only ask the Authors to clarify the definition of seniors. Please
make the references to other studies that discuss the age of seniors.

Ad 4. This part has not been properly prepared by the Authors. The Discussion is the section
of the article in which the Authors critically confront the results of their own research with the
achievements of other researchers. The article lacks such a comparison of the results of own
analyses with the findings of other studies. Based on the results of the analyses conducted by
the means of a survey, the Authors should verify whether the results of their research are
similar or perhaps different to what is presented in the literature.

Ad 5. On page 8, the Authors have placed a questionnaire (in a descriptive form). I suggest
presenting the questionnaire in the form of a table, it will be more understandable for the
reader. I leave to the Authors the decision whether to keep such a table in the present place
in the article or move it to its Appendix. One more issue regarding the questionnaire. I would
like to ask the Authors to comment on what basis the coordinates of passive variables were
determined.

Ad 6. The structure of the article is not discussed in the Introduction. Please add this part.

Minor linguistic/stylistic errors can be found, e.g.: the sentence: In the first analysis are three
describing tourist and physical activity (Q1, Q2, Q5) and four demographic variables.”

I recommend linguistic proofreading to eliminate any linguistic imperfections.

Overall, I believe that this article is the work of high quality and deserves, after the suggested
modifications have been made, to be published.

A re-review is not necessary.

Author Response

Response to Reviewer 2 Comments

Dear Reviewers and Dear Editors,

Thank you very much for all, very important remarks. We are submitting the article in the track changes mode to make it easier to identify all the items that have been improved, moved, or added to the original version of the article. Since the reviewers indicated various areas that we should improve, we tried to create a coherent version containing as many valuable suggestions from the reviewers as possible. We are very grateful for such detailed comments.

The article presented for review, entitled “Tourism-related needs in the context of seniors’

living and social conditions” is an interesting and valuable work.

Dear Reviewer, thank you for your initial opinion and detailed comments. We will try to address every problematic issue.

In my review, I would like to highlight the following issues:

  1. No clearly defined purpose of the article.
  2. Overview of data sources.
  3. In the part concerning the literature review, I suggest distinguishing two sub-section -

one related to Active Ageing and the other related to the determinants of senior

tourism.

  1. The reformulation of discussion.
  2. Including the questionnaire in the Appendix.
  3. Presenting the article structure.

The issues are discussed in more detail below.

Ad 1. In the article presented for review, two closely related, but substantially different

threads can be distinguished. The first one is based on Eurostat data and is related to

international comparisons in the area of factors that limit the participation of seniors in

tourism. The other one concerns the discussion of the results of research carried out on a

sample of seniors in Poland based on a survey questionnaire. Bearing this in mind, I

recommend that the Authors present the main purpose of the article in the Introduction (also

in the Abstract). Additionally, due to the above-described two-threading nature of the

presented analyses, I consider it justified to support the main purpose by providing two

specific aims. Thus, the purpose related to the research carried out by the Authors, which is

included in sub-section 4.2, can be used as one of the specific aims. I also recommend that the

Authors should develop the sentence on page 2, emphasising areas related to the research

topic and the scope of the journal’s special issue.

Thank you very much for this very important remark. We admit that in the abstract, the first two sentences are too descriptive of the purpose of our study. We propose to change it to:

Significant changes are taking place in the structure of tourism participants. Due to the ageing of societies, the tourism sector has to respond to the increasing tourist activity of seniors. The main aim of our research was the recognition of the needs of senior tourists from selected regions of Poland, considering their health and financial situation as well as their physical activity. The study shows how to combine the knowledge of assumptions of active ageing with the actual views of senior tourists on tourism and active leisure. An additional objective was to determine the reasons why seniors gave up tourism and to compare the reasons why seniors from selected regions of Poland and seniors from other European countries did not participate in tourism.

We have expanded the next sentence: „In studies related to active ageing, the authors repeatedly emphasise the growing importance of this phenomenon and its impact on various areas of the economy [2,3,9-12].” Currently, it is an excerpt:

In studies related to active ageing, the authors repeatedly emphasise the growing importance of this phenomenon. Adding to this the significance of individual activity understood as senior tourism, this phenomenon becomes multi-faceted. Two areas of influence by senior tourism can be identified: individual and socio-economic. In the first area, factors directly influencing the elderly as a social actor should be identified. In the other one, factors influencing the functioning of the various areas of the economy should be identified [2,3,9-12]. These are, generally, the health system and economic growth, the labour market and the growth of the tourism market. These issues are synergistic (they are in synergy and function indissoluble); the more tourism is considered the more important the industry becomes [2, 13, 14].

Ad 2. Regarding Eurostat, please explain why the data from 2016 and 2019 are compared (it

is sufficient to explain that these are the most recent data available from Eurostat at the time

of writing the article). I would also like to ask for a few additional sentences to explain what

research the data come from, how they are collected, the frequency with which it is carried

out, and what data are missing in Eurostat databases and the extent to which it influenced the

analyses presented by the Authors in the article. The Authors take up this thread partially in

section 3 and partially in sub-section 4.1, it would be worthwhile to collect it as a whole and

include it in section 4.1.

In our study, we considered a significant indication that it is not only seniors in Poland who do not participate in tourism, but also seniors from other parts of Europe. We were also interested in the fundamental problems that affect seniors' withdrawal from tourism. For this reason, we have devoted part of our article to the non-participation of seniors in tourism. We take it as a background to our study. Perhaps we have articulated this aim of our research too weakly. The data that enables the results obtained in the EU countries to be compared is available in Eurostat. However, this data is aggregates and not individual data. Data on non-participation in tourism has been collected so far for 2013, 2016 and 2019. The 2019 data was released in July 2021. So with a lot of delay. However, the important thing about Eurostat data is that it is representative data (as shown by the metadata file). The surveyed population is people aged 15 and more. As we wrote: „The age range is predetermined. With reference to the above-mentioned groups of seniors, it was also possible to check the reasons for not participating in tourism among people aged 55-64. However, we decided that these people, compared to the elderly, are not so much at risk of being excluded from tourism because more often they remain professionally active and their health is better.” In 2016, according to Eurostat data, 48 million people aged 65+ did not participate in tourism, and in 2019 - 44 million people of this age. They constituted 50% and 43% of people in this age group in the European Union, respectively. In our analysis, we did not take into account the data from 2013 due to the large time gap to the survey we conducted.

Eurostat data on non-participation in tourism is complete data in the area established by Eurostat. A drawback of the data available in Eurostat is the inability to check the situation of non-participation in tourism by seniors, broken down into different demographic groups depending on education, place of residence, etc.

We propose to add the following passage in Chapter 3:

In our study, we considered a significant indication that it is not only seniors in Poland who do not participate in tourism, but also seniors from other parts of Europe. To indicate the fundamental reasons for the resignation of European seniors from tourism, we used Eurostat data from 2016 and 2019. We took the analysis of the non-participation of seniors in tourism as the background to the main study. The data available in Eurostat is aggregates and not individual data. Data on non-participation in tourism has been collected so far for 2013, 2016 and 2019. The 2019 data was released in July 2021. So with a lot of delay. Eurostat data is representative (as shown by the metadata file). The surveyed population is people aged 15 and over. The age range is predetermined. Based on this data, it is possible to identify the most common reasons for people who do not participate in tourism and who are over 65 years of age. With regard to the above-mentioned groups of seniors, it was also possible to examine the reasons for non-participation in tourism among those aged 55-64. However, we decided that these people, compared to the elderly, are not so much at risk of being excluded from tourism because more often they remain professionally active and their health is better. In 2016, according to Eurostat data, 48 million people aged 65+ did not participate in tourism, and in 2019 - 44 million people of this age. They constituted 50% and 43% of people in this age group in the European Union, respectively. In our analysis, we did not take into account the data from 2013 due to the large time gap to the survey we conducted.

Eurostat data on non-participation in tourism is complete data in the area established by Eurostat. A drawback of the data available in Eurostat is the inability to check the situation of non-participation in tourism by seniors, broken down into different demographic groups depending on education, place of residence, etc. The second source of data that we used allowed us to identify indicate the reasons for non-participation in tourism depending on various demographic characteristics.

Questionnaire survey

Defining the geographical scope of the research - page 5, the Authors point to “three regions

of Silesia in Poland (Lower, Upper and Opole).” I suggest that with regard to the geographical

area, the division into NUTS classification units applicable in the EU should be used, or if it is

not possible, the Authors should refer to them in some way. The research results will then be

more understandable to a foreign reader. I would also like that the Authors provide an

explanation to the choice of the snowball technique.

Generally, due to two different data sources, I suggest that their characteristics should be

included in sub-section 4.1 (Eurostat) and section 4.2 (questionnaire survey).

In section 3, I suggest only a general comment that data sources, due to their diversity, will be

discussed in the following sub-sections of the article, respectively in sub-section 4.1 (data

obtained from the Eurostat database) and 4.2 (data collected on the basis of the Authors' own

questionnaire, including the data collection method, the sample size, the geographical scope,

etc.). It will surely be more understandable for the reader then.

We propose to add the following passage in Chapter 3

In 2020, just before the outbreak of the Covid-19 pandemic, we conducted a survey among seniors. The respondents were selected as a 65 years and older sample. The survey was conducted in three regions of Silesia in Poland (NUTS2): Dolnośląskie (PL51), Opolskie (PL52), Śląskie (PL22) - Fig. 1. It is the south-west part of Poland which borders Germany and the Czech Republic. There are two airports in this region serving flights to most European countries and airports in Poland. There is also a very dense railway network. It is possible to connect with the countries of the south and west of Europe and with many attractive tourist destinations in Poland. In addition, the A4 and A1 motorways run through these three regions. A snowball technique was applied when choosing the next interviewee. The choice of this form of research was based on several crucial aspects. Older people are reluctant to fill in questionnaires and participate in a study. The number of questions and the complexity of the questionnaire may also contribute to their reluctance towards a survey. The snowball technique reduces the respondents' fear of the survey. Sixty-eight respondents took part in the survey, one of whom refused to give answers. The structure of the respondents is presented in Table 1. The number of men participating in the survey reflects the changes in the demographic structure with the ageing of the population.

Ad 3.

Please divide the section on the literature review as suggested earlier. I do not assume the

extensive expansion of this section with new elements added, but mainly the identification of

thematically coherent points. I only ask the Authors to clarify the definition of seniors. Please

make the references to other studies that discuss the age of seniors.

Thank you very much for this suggestion. Adding sections to our work will contribute to its greater transparency. We propose the following sections in Chapter 2:

Active ageing and quality of life

Seniors ’age determination

Seniors ’exclusion

Ad 4. This part has not been properly prepared by the Authors. The Discussion is the section

of the article in which the Authors critically confront the results of their own research with the

achievements of other researchers. The article lacks such a comparison of the results of own

analyses with the findings of other studies. Based on the results of the analyses conducted by

the means of a survey, the Authors should verify whether the results of their research are

similar or perhaps different to what is presented in the literature.

As the reviewer rightly notes, in the Discussion section, there were no references to other studies of tourism of seniors living in southwest Poland or the exclusion of seniors from tourism. This is due to two reasons. First of all, we thought that the article was already very extensive. That is why we gave up such a comparison. We are happy to add an excerpt referring to such studies. Hence, we included a new paragraph, in the discussion chapter:

In the literature, some reasons for travelling are given by seniors. They focus on the fulfilment of needs and motives for action such as visiting family or friends, social contacts and participation in social life, which are closely related. When choosing a travel destination, seniors pay attention to climate and weather, especially a pleasant air temperature [42]. Grześkowiak et al. [1] and Liew et al. [4] indicate the importance of safety, hygiene and cleanliness of the accommodation. A certain attempt at classifying destination choices is grouping them into universal and specific. The universal or key reasons are: unique natural [11, 13], scenic resources [11, 13], comfortable climate [11], safety of tourism attractions [11], relaxation, well-being, socialisation and self-esteem [13]. Specific factors include barrier-free: public transportation, accommodation and facilities along travel routes [4, 11, 34]. Factors such as self-development and relaxation [13] as well as physical activity [34] are evidence of the changing perception of tourism by seniors towards more active pursuits with a focus on health and fitness.

The types of tourism chosen by seniors also influence the motives for tourism choices. Mayer [43] indicates that the most popular among people aged 65 and over are leisure, cognitive, health (spa, fitness and wellness), religious (religious and cognitive, cognitive and pilgrimage) and ethnic (sentimental) tourism. McCabe and Qiao [44] distinguish social tourism popular among seniors struggling with disabilities and age-related limitations. This type of tourism supports the pursuit of happiness, satisfaction, health and social inclusion. Tourism of seniors is social tourism because seniors are a group that may be excluded from participation in regular tourism due to the limitation or difficulty of access [13, 45].

However, these studies had a different scope.

Ad 5. On page 8, the Authors have placed a questionnaire (in a descriptive form). I suggest

presenting the questionnaire in the form of a table, it will be more understandable for the

reader. I leave to the Authors the decision whether to keep such a table in the present place

in the article or move it to its Appendix. One more issue regarding the questionnaire. I would

like to ask the Authors to comment on what basis the coordinates of passive variables were

determined.

Many thanks to the reviewer for this her their suggestion. Indeed, such an arrangement will make the argument more transparent.

In Annex 1, we are going to add a questionnaire translated into English, which contains both a cover letter and questions, along with encoding the answers.

Considering all the reviewers' suggestions, we decided to introduce changes to 4.2. We moved the objectives, the description of the questionnaire and the sample characteristics to chapters 1 and 2, respectively. At the beginning of chapter 4.2, we indicated which parts of the questionnaire are devoted to the research areas: physical activity, willingness to participate in tourism and tourist behaviour:

The tourist behaviours of Polish seniors were examined based on the questionnaire presented in Appendix 1.

A structured questionnaire (Appendix 1) was developed to identify the key characteristics of seniors' perceptions and involvement in tourism and their physical activity. The questionnaire consisted of three main sections that examined: physical activity and the willingness of seniors to travel (Q1-Q6), then the reasons for not participating in tourism (Q7-Q9), and finally the behaviours of senior tourists (Q10-Q13). The questionnaire ends with socio-demographic questions. This study adopts mainly correspondence analysis to investigate the characteristics of seniors as travellers.

We propose to add the following passage in Chapter 3:

Passive variables (additional points) illustrate the information that is not presented in the analysed interactions of the categories of active variables. The coordinates of passive variables are determined on the basis of singular values and singular vectors determined during the analysis of active variables. The inclusion of passive points in the analysis occurs only after performing all calculations for the variables whose relations are examined (active variables).

Ad 6. The structure of the article is not discussed in the Introduction. Please add this part.

Minor linguistic/stylistic errors can be found, e.g.: the sentence: “In the first analysis are three

describing tourist and physical activity (Q1, Q2, Q5) and four demographic variables.”

We would also like to thank the reviewer for pointing out this error. We are going to replace this sentence with a new one:

In the first analysis are three variables describing tourist and physical activity (Q1, Q2, Q5) and five demographic variables (Gender, Age, Marital status, Level of education, Place of residence).

I recommend linguistic proofreading to eliminate any linguistic imperfections.

As suggested linguistic proofreading done

Overall, I believe that this article is the work of high quality and deserves, after the suggested

modifications have been made, to be published.

A re-review is not necessar

Reviewer 3 Report

I put my findings as under:

Abstract- is very confusing and not written in a standard scientific paper, i.e., motivation, data and methodology, results, conclusions, and implications. Instead, the authors appear to explain the article as they remember things and not obeying a guiding thread. The two methodologies are not explained in the abstract and therefore one does not know by the abstract what is the population of each study. Therefore, the conclusions are very confusing because one refers only to polish seniors and the other to European seniors.

The contribution of the paper to the literature is not well emphasized. The manuscript needs to provide clear research questions and hypotheses in the introduction. The introduction does not include an explanation of the structure of the remaining paper.

Table 1- The title is not informative of the table.

The literature review section is incipient and does not have enough (recent) references. I suggest at least 60 references for the whole paper. Therefore, the discussion section is not well structured, it needs the results from previous studies to put this study’s results into perspective vis-à-vis previous literature.

Material and methods. This section should start with a description of data and its collection, potential problems and its consequences, the sample representation, etc.

The questionnaire designed for senior tourism should have been included as an appendix. Where does it come from? How did the authors validate their questionnaire? I have serious doubts about it. For example, one of the questions is single or non-single. The same goes for education: the authors divide it into higher education and other. There may be a spectrum of other possibilities in between that may affect the respondents’ choices towards tourism!

Data collection protocol should have been well presented in the manuscript, including the choice of the sample, the consent to publish, and ethical issues.

The results should be discussed in a more detailed and critical way.

What are the implications of the results for tourism destinations? 

Author Response

Dear Reviewers and Dear Editors,

Thank you very much for all, very important remarks. We are submitting the article in the track changes mode to make it easier to identify all the items that have been improved, moved, or added to the original version of the article. Since the reviewers indicated various areas that we should improve, we tried to create a coherent version containing as many valuable suggestions from the reviewers as possible. We are very grateful for such detailed comments.

Response to Reviewer 3 Comments

Abstract- is very confusing and not written in a standard scientific paper, i.e., motivation, data and methodology, results, conclusions, and implications. Instead, the authors appear to explain the article as they remember things and not obeying a guiding thread. The two methodologies are not explained in the abstract and therefore one does not know by the abstract what is the population of each study. Therefore, the conclusions are very confusing because one refers only to polish seniors and the other to European seniors.

Thank you very much for this remark. We have revised the summary to include the general premises of the study, the purpose of the study, data sources, methods of analysis, and main conclusions. Unfortunately, the analysis methods cannot be discussed in detail in the summary because the abstract that the journal expects must be very short. We have changed the fragment concerning the aim of the study included in the abstract so that the connection between the two research areas is visible:

Significant changes are taking place in the structure of tourism participants. Due to the ageing of societies, the tourism sector has to respond to the increasing tourist activity of seniors. The main aim of our research was the recognition of the needs of senior tourists from selected regions of Po-land, considering their health and financial situation as well as their physical activity. The study shows how to combine the knowledge of assumptions of active ageing with the actual views of senior tourists on tourism and active leisure. An additional objective was to determine the reasons why seniors gave up tourism and to compare the reasons why seniors from selected regions of Poland and seniors from other European countries did not participate in tourism. Based on Eurostat data, we identify the most common reasons for people not participating in tourism who are over 65 years of age. In 2020 we surveyed seniors. The respondents were selected as 65 years and older sample. In order to compare countries due to exclusion and non-participation of seniors in tourism, the results classification was used. To analyse the touristic behaviours of Polish seniors, we used correspondence analysis. As indicated by analysing the reasons for the non-participation of Europeans aged 65 and over in tourism, in most countries, financial and health reasons are ranked first or second in 2016 and 2019. In a survey of Polish seniors, except for the financial reasons responsible for non-participation in tourism, an additional obstacle was the language barrier in foreign tourism. The analysis of physical and tourist activity showed that non-participation in tourism is associated with low physical activity. Women reported that they were satisfied with their financial independence and most often used the opportunity of short-term tourism. The people who are fully or largely involved in organising their trips also willingly change their locations during their next travels.

The contribution of the paper to the literature is not well emphasized. The manuscript needs to provide clear research questions and hypotheses in the introduction. The introduction does not include an explanation of the structure of the remaining paper.

Thank you very much for pointing out this mistake. The addition of sections comparing our study with the previous research in analysing the expectations, habits and tourist opportunities of seniors in the south-west and west of Poland and their exclusion from tourism will highlight our contribution to the literature in this area. We are adding the following excerpt in Chapter 5:

In the literature, some reasons for travelling are given by seniors. They focus on the fulfilment of needs and motives for action such as visiting family or friends, social contacts and participation in social life, which are closely related. When choosing a travel destination, seniors pay attention to climate and weather, especially a pleasant air temperature [42]. Grześkowiak et al. [1] and Liew et al. [4] indicate the importance of safety, hygiene and cleanliness of the accommodation. A certain attempt at classifying destination choices is grouping them into universal and specific. The universal or key reasons are: unique natural [11, 13], scenic resources [11, 13], comfortable climate [11], safety of tourism attractions [11], relaxation, well-being, socialisation and self-esteem [13]. Specific factors include barrier-free: public transportation, accommodation and facilities along travel routes [4, 11, 34]. Factors such as self-development and relaxation [13] as well as physical activity [34] are evidence of the changing perception of tourism by seniors towards more active pursuits with a focus on health and fitness.

The types of tourism chosen by seniors also influence the motives for tourism choices. Mayer [43] indicates that the most popular among people aged 65 and over are leisure, cognitive, health (spa, fitness and wellness), religious (religious and cognitive, cognitive and pilgrimage) and ethnic (sentimental) tourism. McCabe and Qiao [44] distinguish social tourism popular among seniors struggling with disabilities and age-related limitations. This type of tourism supports the pursuit of happiness, satisfaction, health and social inclusion. Tourism of seniors is social tourism because seniors are a group that may be excluded from participation in regular tourism due to the limitation or difficulty of access [13, 45].

Considering all the reviewers' suggestions, we decided to introduce changes to 4.2. We moved the objectives a research questions, the description of the questionnaire and the sample characteristics to chapters 1 and 2, respectively.

At the end of the introduction, a description of the article structure has been added

Table 1- The title is not informative of the table.

After considering all the reviewers' comments, the article's structure has been modified. Table 1 in the original version of the article is now number 2. Its title reflects the purpose of the analysis we have performed. As we noted regarding the table (currently 2): The first group includes countries where a given cause is indicated as the most important, and the fourth group is the least important.

The literature review section is incipient and does not have enough (recent) references. I suggest at least 60 references for the whole paper. Therefore, the discussion section is not well structured, it needs the results from previous studies to put this study’s results into perspective vis-à-vis previous literature.

Thank you for your suggestion to increase the number of references. Unfortunately, while reviewing the literature, we encountered two problems. One was repeated references to theoretical issues by the same authors, which, unfortunately, contributed nothing new. The other one concerns the target group of respondents. Research on seniors in southwestern and southern Poland is not frequent, nor is research focused on the tourist needs of seniors from other European countries. Most studies on the tourist needs of seniors are carried out in Asian countries. However, there are significant differences between these target groups. Only the needs of people entering senior age and living in the European Union are comparable.

In our article, we have provided a fragment with reference and comparison to the research conducted so far in Poland in the Dolnośląskie region, ie PL51, according to NUTS2.

We have enriched our article with the following literature items:

  1. Grzelak, M.M.; Roszko-Wójtowicz, E. Tourist attractiveness of voivodeships in Poland in the light of selected indicators: a dynamic approach. Economic Annals-XXI 2020, 184(7-8), 161-177. doi: https://doi.org/10.21003/ea.V184-14.
  2. European Commission. Assessment of the Europe 2020 strategy joint report of the Employment Committee (EMCO) and Social Protection Committee (SPC). Publications office of the European Union, Luxembourg, 2019. Available online: https://ec.europa.eu/social/main.jsp? langId=en&catId=1063&furtherNews=yes&newsId=9487 (accessed on 05 August 2022).
  3. Begg, I. Europe 2020 and employment. In Europe 2020 – A Promising Strategy?, Bongardt A., Torres, F. Intereconomics, 45, 2010; pp. 146-51. DOI:10.1007/s10272-010-0332-9. 2010.
  4. European Commission. Communication from the Commission to the European Parliament, the Council, the European Economic and Social Committee and the Committee of the Regions. The European Platform Against Poverty and Social Exclusion: A European framework for social and territorial cohesion. COM(2010) 758 final. Available online: https://op.europa.eu/en/publication-detail/-/publication/16456b4c-211e-434f-9884-a11b6d6a7f79/language-en/format-PDF/source-search (accessed on 05 August 2022).
  5. Przybysz, K.; Stanimir A.; Wasiak, M. Subjective Assessment of Seniors on the Phenomenon of Discrimination: Analysis Against the Background of the Europe 2020 Strategy Implementation. European Research Studies Journal 2021, Vol. XXIV Special Issue 1, 810-35. DOI: 10.35808/ersj/2075.
  6. Steiger, R.; Abegg, B.; Jänicke, L. Rain, rain, go away, come again another day. Weather preferences of summer tourists in mountain environments. Atmosphere 2016, 7, 5, 63. DOI: 10.3390/atmos7050063.
  7. Meyer, B. Pozostałe formy obsługi ruchu turystycznego. In Gospodarka turystyczna, Panasiuk, A. (ed.). Wydawnictwo Naukowe PWN: Warsaw, Poland, 2008; pp. 164-184
  8. McCabe, S.; Qiao, G. A review of research into social tourism: Launching the Annals of Tourism Research Curated Collection on Social Tourism. Annals of Tourism Research 2020, 85, 103103. DOI:10.1016/j.annals.2020.103103.
  9. Markiewicz-Patkowska. J.; Pytel, S.; Widawski, K.; OleÅ›niewicz, P. Turystyka senioralna w kontekÅ›cie sytuacji materialnej polskich emerytów. Ekonomiczne problemy turystyki 2018, 2, 42. DOI:10.18276/ept.2018.2.42-10.
  10. OleÅ›niewicz, P.; Widawski, K. Motywy podejmowania aktywnoÅ›ci turystycznej przez osoby starsze ze Stowarzyszenia Promocji Sportu FAN. Rozprawy Naukowe AWF we WrocÅ‚awiu 2015, 51, 15–24.
  11. OleÅ›niewicz, P.; Markiewicz-Patkowska, J.; Widawski, K. Senior tourism on the example of members of the Association for the Promotion of Sports “Fan” in Wroclaw. In Proceedings of the 10th International Conference on Kinantrhopology, Brno, Czech Republic, November 18–20, pp. 259–270.
  12. Zielińska-Szczepkowska, J. What Are the Needs of Senior Tourists? Evidence from Remote Regions of Europe. Economies 2021, 9: 148. https://doi.org/10.3390/economies9040148.

49 Patterson, I.; Balderas-Cejudo, A.; Pegg, S. Tourism preferences of seniors and their impact on healthy ageing. Anatolia 2021, 32:4, 553-564, DOI: 10.1080/13032917.2021.1999753.

  1. Markiewicz-Patkowska, J.; Widawski, K.; Oleśniewicz, P. Conditionings of the educational function of senior tourism on the example of seniors from Wroclaw, Poland. Journal of International Scientific Publications 2017, 15, 402-11.
  2. Markiewicz-Patkowska, J.; Widawski, K.; Oleśniewicz, P. Selected conditions of the senior tourism functioning on the example of Lower Silesia. Studia sportive 2017, 11, 2, 106-14.
  3. Alén, E.; Nicolau, J.L.; Losada, N.; Domínguez, T. Determinant factors of senior tourists’ length of stay. Annals of Tourism Research 2014, 49, 2014, 19-32. https://doi.org/10.1016/j.annals.2014.08.002.

Material and methods. This section should start with a description of data and its collection, potential problems and its consequences, the sample representation, etc.

Thank you very much for this remark. In Chapter 3 we propose the following structure of this part of the article: description of data (both sources), its collection, and used methods.

The questionnaire designed for senior tourism should have been included as an appendix.

Many thanks to the reviewer for this her their suggestion. Indeed, such an arrangement will make the argument more transparent.

In Annex 1, we are going to add a questionnaire translated into English, which contains encoding the answers.

 Where does it come from? How did the authors validate their questionnaire? I have serious doubts about it. For example, one of the questions is single or non-single. The same goes for education: the authors divide it into higher education and other. There may be a spectrum of other possibilities in between that may affect the respondents’ choices towards tourism!

Thank you very much for this question. As indicated in the article, the snowball method was used to collect the data. This was due to several reasons. First, respondents aged 65+ do not willingly answer postal, telephone or Internet surveys. The latter method is a perfect form of data collection for young people, even in a long survey, but for older people in Poland, this method is practically unavailable.

The second problem was that the questions had an elaborate structure. Thus, they required verbal instructions. In addition, older people are more willing to fill in the questionnaire, i.e. they are not embarrassed when they know that someone they know also participates in the survey. This way of distributing the questionnaire has social advantages.

The reviewer rightly pointed out that dividing respondents by education into higher education and other is questionable. We agree that in between, there may be a whole range of other possibilities that could influence respondents' choices about tourism. However, please remember that the possibility of obtaining higher education by people aged 65+ and older (especially 75+) in Poland was not as common as today. Thus, people with higher education had the opportunity to get a better job with a guaranteed social benefit in the form of a retirement pension, thus constituting a separate social group throughout their active lives.

Data collection protocol should have been well presented in the manuscript, including the choice of the sample, the consent to publish, and ethical issues.

We agree with the reviewer. Ethical issues are presented at the end of the article, before References.

The results should be discussed in a more detailed and critical way.

In Chapter 5, a reference to other studies, to the scope of the study and the results obtained was added:

Research conducted among Polish seniors indicates that they do not travel for fi-nancial reasons [45-48] or for health reasons [46-48]. This conclusion is in line with the results of our study. However, the survey we conducted among selected seniors showed yet another reason which was a language barrier. The authors of many studies indicate that seniors choose domestic stays, which they organise themselves or with the help of travel agencies [45-48]. Seniors in other countries do the same, making ad-ditional use of the Internet [49]. The purpose of trips by Polish seniors is leisure and family reunions [50, 51]. Leisure tourism also combines cognitive values [46-48, 50, 51]. The seniors participating in our study gave us similar feedback. However, we also in-dicated differences between seniors resulting from socio-demographic characteristics. Among other things, an important observation was that active people make active tourists

What are the implications of the results for tourism destinations?

Thank you for the opportunity to add such a request. An appropriate excerpt was added to the article in Chapter 6:

We believe that if you want to develop the tourism industry, you should pay attention to the availability of the tourist offer for seniors and modify it to their needs. Another aspect of developing opportunities for senior citizens to participate in tourism abroad is to encourage them to learn foreign languages. At the same time, it should be remembered that the offer directed to seniors should be as attractive as the one directed to young people. No shortcomings of age or physical condition should be emphasised. Adapting the tourist offer to the requirements of seniors has been discussed many times in the literature [46-48, 54]. The tourism industry cannot determine the needs of seniors by itself; it can only do so with significant cooperation with seniors and constant monitoring of their needs and expectations.

Reviewer 4 Report

I begin by congratulating the authors for their scientific approach, especially regarding the chosen topic.

Tourism for the elderly primarily has a huge impact on the health of the elderly, as well as reducing the impact of various negative factors affecting the psyche and the state of the body as a whole. Tourism for the elderly, due to its characteristics, provides constant movement, which helps to eliminate feelings of oppression, stress, loss of confidence in their own strength.

The tourist agencies would say that exploring and discovering the world is possible at any age. We could argue that older people know this very well and find time to travel when they become retired. Generally, retirees go on holiday with regular travel agencies or take advantage of offers dedicated specifically to them. Contrary to popular belief, holidays for the elderly are not just offers for holidays related to rehabilitation or sanatoria. Today's senior citizen is significantly different from 10 or 20 years ago. They are up to date with mobile phone and internet assistance and have no fear of distant travel. Again, this could be another marketing line, but the reality is totally different since there are a lot of financial and health reasons, as the authors also highlight.

I want to believe that there really is a chance for older people to practice tourism. Maybe in the highly developed countries things are different, but for central and south-eastern Europe, I think things are mostly as the authors report.

Older people are somehow forgotten by travel agencies, as there are no clear and honest offers for them, although at European level their number is increasing.

The authors also talk about their exclusion, both financially and in terms of health, and I agree with them.

I hope that this article will spark the interest of other researchers in this field and that we will succeed in clinching this tourism market for the elderly. We would all like to move in the right, honest and healthy direction.

Although the article has some shortcomings, allow me to mention just few of them here (I am confident that the other reviewers will do more), so as not to overshadow the joy of having such a simple approach to this phenomenon, specific to Europe in particular.

Introduction - The introduction is well written, cursive and easy to understand; but quite short and there is a need for a better connection between the idea of Tourism-related needs and the living and social conditions. I consider that in this section should be briefly presented some information about Silesia Province, since it is the area to which the entire material refers and it`s also one of the richest regions in Poland, well connected with Germany and Czech Republic. Perhaps it would be beneficial for the material to add a map

Literature review - although the literature review section is long enough, I suggest a better structure, maybe divided into subchapters to make it easier to follow. I suggest increasing the number of cited works.

Methodology - the presentation of the working methodology…. This is very briefly presented and is based mainly on the exposition of the collection of data and questionnaires. From my point of view, the methodology should also include in detail the way in which the data collected from the respondents were filtered and analyzed, what algorithms were used and how they were used to obtain the final results, what software was used and for what purpose. A well-written section of materials and methods should allow readers to repeat the study themselves, but here it is too vaguely presented to be able to do so.

Discussion and Conclusion - a better exposition would be if these chapters were separated into two: Conclusions, Discussions. This is due to the fact that the results presented in the previous chapter are quite vaguely discussed there. Therefore, the article needs a chapter in which all those analysis and data obtained to be discussed and related to the final purpose of the study.

Author Response

Dear Reviewers and Dear Editors,

Thank you very much for all, very important remarks. We are submitting the article in the track changes mode to make it easier to identify all the items that have been improved, moved, or added to the original version of the article. Since the reviewers indicated various areas that we should improve, we tried to create a coherent version containing as many valuable suggestions from the reviewers as possible. We are very grateful for such detailed comments.

Response to Reviewer 3 Comments

I begin by congratulating the authors for their scientific approach, especially regarding the chosen topic.

Tourism for the elderly primarily has a huge impact on the health of the elderly, as well as reducing the impact of various negative factors affecting the psyche and the state of the body as a whole. Tourism for the elderly, due to its characteristics, provides constant movement, which helps to eliminate feelings of oppression, stress, loss of confidence in their own strength.

The tourist agencies would say that exploring and discovering the world is possible at any age. We could argue that older people know this very well and find time to travel when they become retired. Generally, retirees go on holiday with regular travel agencies or take advantage of offers dedicated specifically to them. Contrary to popular belief, holidays for the elderly are not just offers for holidays related to rehabilitation or sanatoria. Today's senior citizen is significantly different from 10 or 20 years ago. They are up to date with mobile phone and internet assistance and have no fear of distant travel. Again, this could be another marketing line, but the reality is totally different since there are a lot of financial and health reasons, as the authors also highlight.

I want to believe that there really is a chance for older people to practice tourism. Maybe in the highly developed countries things are different, but for central and south-eastern Europe, I think things are mostly as the authors report.

Older people are somehow forgotten by travel agencies, as there are no clear and honest offers for them, although at European level their number is increasing.

The authors also talk about their exclusion, both financially and in terms of health, and I agree with them.

I hope that this article will spark the interest of other researchers in this field and that we will succeed in clinching this tourism market for the elderly. We would all like to move in the right, honest and healthy direction.

We would like to thank the reviewer for evaluating our article.

Although the article has some shortcomings, allow me to mention just few of them here (I am confident that the other reviewers will do more), so as not to overshadow the joy of having such a simple approach to this phenomenon, specific to Europe in particular.

Introduction - The introduction is well written, cursive and easy to understand; but quite short and there is a need for a better connection between the idea of Tourism-related needs and the living and social conditions. I consider that in this section should be briefly presented some information about Silesia Province, since it is the area to which the entire material refers and it`s also one of the richest regions in Poland, well connected with Germany and Czech Republic. Perhaps it would be beneficial for the material to add a map

We would like to thank the Reviewer for the suggestion to enter a description of the region. In the article, we add a relevant paragraph and a map. Taking into account the recommendations of all the reviewers, we add this paragraph in part 3. In the description of the region, we focus on the availability of transport infrastructure.

In 2020, just before the outbreak of the Covid-19 pandemic, we conducted a survey among seniors. The respondents were selected as a 65 years and older sample. The survey was conducted in three regions of Silesia in Poland (NUTS2): Dolnośląskie (PL51), Opolskie (PL52), Śląskie (PL22) - Fig. 1. It is the south-west part of Poland which borders Germany and the Czech Republic. There are two airports in this region serving flights to most European countries and airports in Poland. There is also a very dense railway network. It is possible to connect with the countries of the south and west of Europe and with many attractive tourist destinations in Poland. In addition, the A4 and A1 motorways run through these three regions. A snowball technique was applied when choosing the next interviewee. The choice of this form of research was based on several crucial aspects. Older people are reluctant to fill in questionnaires and participate in a study. The number of questions and the complexity of the questionnaire may also contribute to their reluctance towards a survey. The snowball technique reduces the respondents' fear of the survey. Sixty-eight respondents took part in the survey, one of whom refused to give answers. The structure of the respondents is presented in Table 1. The number of men participating in the survey reflects the changes in the demographic structure with the ageing of the population.

Literature review - although the literature review section is long enough, I suggest a better structure, maybe divided into subchapters to make it easier to follow. I suggest increasing the number of cited works.

We have enriched our article with the following literature items:

  1. Grzelak, M.M.; Roszko-Wójtowicz, E. Tourist attractiveness of voivodeships in Poland in the light of selected indicators: a dynamic approach. Economic Annals-XXI 2020, 184(7-8), 161-177. doi: https://doi.org/10.21003/ea.V184-14.
  2. European Commission. Assessment of the Europe 2020 strategy joint report of the Employment Committee (EMCO) and Social Protection Committee (SPC). Publications office of the European Union, Luxembourg, 2019. Available online: https://ec.europa.eu/social/main.jsp? langId=en&catId=1063&furtherNews=yes&newsId=9487 (accessed on 05 August 2022).
  3. Begg, I. Europe 2020 and employment. In Europe 2020 – A Promising Strategy?, Bongardt A., Torres, F. Intereconomics, 45, 2010; pp. 146-51. DOI:10.1007/s10272-010-0332-9. 2010.
  4. European Commission. Communication from the Commission to the European Parliament, the Council, the European Economic and Social Committee and the Committee of the Regions. The European Platform Against Poverty and Social Exclusion: A European framework for social and territorial cohesion. COM(2010) 758 final. Available online: https://op.europa.eu/en/publication-detail/-/publication/16456b4c-211e-434f-9884-a11b6d6a7f79/language-en/format-PDF/source-search (accessed on 05 August 2022).
  5. Przybysz, K.; Stanimir A.; Wasiak, M. Subjective Assessment of Seniors on the Phenomenon of Discrimination: Analysis Against the Background of the Europe 2020 Strategy Implementation. European Research Studies Journal 2021, Vol. XXIV Special Issue 1, 810-35. DOI: 10.35808/ersj/2075.
  6. Steiger, R.; Abegg, B.; Jänicke, L. Rain, rain, go away, come again another day. Weather preferences of summer tourists in mountain environments. Atmosphere 2016, 7, 5, 63. DOI: 10.3390/atmos7050063.
  7. Meyer, B. Pozostałe formy obsługi ruchu turystycznego. In Gospodarka turystyczna, Panasiuk, A. (ed.). Wydawnictwo Naukowe PWN: Warsaw, Poland, 2008; pp. 164-184
  8. McCabe, S.; Qiao, G. A review of research into social tourism: Launching the Annals of Tourism Research Curated Collection on Social Tourism. Annals of Tourism Research 2020, 85, 103103. DOI:10.1016/j.annals.2020.103103.
  9. Markiewicz-Patkowska. J.; Pytel, S.; Widawski, K.; OleÅ›niewicz, P. Turystyka senioralna w kontekÅ›cie sytuacji materialnej polskich emerytów. Ekonomiczne problemy turystyki 2018, 2, 42. DOI:10.18276/ept.2018.2.42-10.
  10. OleÅ›niewicz, P.; Widawski, K. Motywy podejmowania aktywnoÅ›ci turystycznej przez osoby starsze ze Stowarzyszenia Promocji Sportu FAN. Rozprawy Naukowe AWF we WrocÅ‚awiu 2015, 51, 15–24.
  11. OleÅ›niewicz, P.; Markiewicz-Patkowska, J.; Widawski, K. Senior tourism on the example of members of the Association for the Promotion of Sports “Fan” in Wroclaw. In Proceedings of the 10th International Conference on Kinantrhopology, Brno, Czech Republic, November 18–20, pp. 259–270.
  12. Zielińska-Szczepkowska, J. What Are the Needs of Senior Tourists? Evidence from Remote Regions of Europe. Economies 2021, 9: 148. https://doi.org/10.3390/economies9040148.

49 Patterson, I.; Balderas-Cejudo, A.; Pegg, S. Tourism preferences of seniors and their impact on healthy ageing. Anatolia 2021, 32:4, 553-564, DOI: 10.1080/13032917.2021.1999753.

  1. Markiewicz-Patkowska, J.; Widawski, K.; Oleśniewicz, P. Conditionings of the educational function of senior tourism on the example of seniors from Wroclaw, Poland. Journal of International Scientific Publications 2017, 15, 402-11.
  2. Markiewicz-Patkowska, J.; Widawski, K.; Oleśniewicz, P. Selected conditions of the senior tourism functioning on the example of Lower Silesia. Studia sportive 2017, 11, 2, 106-14.
  3. Alén, E.; Nicolau, J.L.; Losada, N.; Domínguez, T. Determinant factors of senior tourists’ length of stay. Annals of Tourism Research 2014, 49, 2014, 19-32. https://doi.org/10.1016/j.annals.2014.08.002.

Methodology - the presentation of the working methodology…. This is very briefly presented and is based mainly on the exposition of the collection of data and questionnaires. From my point of view, the methodology should also include in detail the way in which the data collected from the respondents were filtered and analyzed, what algorithms were used and how they were used to obtain the final results, what software was used and for what purpose. A well-written section of materials and methods should allow readers to repeat the study themselves, but here it is too vaguely presented to be able to do so.

Thank you very much for this question and suggestions on how to address the problem. As indicated in the article, the snowball method was used to collect the data. This was due to several reasons. First, respondents aged 65+ do not willingly answer postal, telephone or Internet surveys. The latter method is a perfect form of data collection for young people, even in a long survey, but for older people in Poland, this method is practically unavailable.

The second problem was that the questions had an elaborate structure. Thus, they required verbal instructions. In addition, older people are more willing to fill in the questionnaire, i.e. they are not embarrassed when they know that someone they know also participates in the survey. This way of distributing the questionnaire has social advantages. In the fragment presenting Eurostat data, descriptions were added to render a more in-depth assessment of this material. The type of software used was indicated (Statistica 13.3) – sources of Figures 2-3.

Discussion and Conclusion - a better exposition would be if these chapters were separated into two: Conclusions, Discussions. This is due to the fact that the results presented in the previous chapter are quite vaguely discussed there. Therefore, the article needs a chapter in which all those analysis and data obtained to be discussed and related to the final purpose of the study.

Thank you very much for your valuable comments in this area. In the Discussion we refer to other studies.

In the literature, some reasons for travelling are given by seniors. They focus on the fulfilment of needs and motives for action such as visiting family or friends, social contacts and participation in social life, which are closely related. When choosing a travel destination, seniors pay attention to climate and weather, especially a pleasant air temperature [42]. Grześkowiak et al. [1] and Liew et al. [4] indicate the importance of safety, hygiene and cleanliness of the accommodation. A certain attempt at classifying destination choices is grouping them into universal and specific. The universal or key reasons are: unique natural [11, 13], scenic resources [11, 13], comfortable climate [11], safety of tourism attractions [11], relaxation, well-being, socialisation and self-esteem [13]. Specific factors include barrier-free: public transportation, accommodation and facilities along travel routes [4, 11,34]. Factors such as self-development and relaxation [13] as well as physical activity [34] are evidence of the changing perception of tourism by seniors towards more active pursuits with a focus on health and fitness.

The types of tourism chosen by seniors also influence the motives for tourism choices. Meyer [43] indicates that the most popular among people aged 65 and over are leisure, cognitive, health (spa, fitness and wellness), religious (religious and cognitive, cognitive and pilgrimage) and ethnic (sentimental) tourism. McCabe and Qiao [44] distinguish social tourism popular among seniors struggling with disabilities and age-related limitations. This type of tourism supports the pursuit of happiness, satisfaction, health and social inclusion. Tourism of seniors is social tourism because seniors are a group that may be excluded from participation in regular tourism due to the limitation or difficulty of access [13, 45].

Research conducted among Polish seniors indicates that they do not travel for financial reasons [45-48] or for health reasons [46-48]. This conclusion is in line with the results of our study. However, the survey we conducted among selected seniors showed yet another reason which was a language barrier. The authors of many studies indicate that seniors choose domestic stays, which they organise themselves or with the help of travel agencies [45-48]. Seniors in other countries do the same, making additional use of the Internet [49]. The purpose of trips by Polish seniors is leisure and family reunions [50, 51]. Leisure tourism also combines cognitive values [46-48, 50, 51]. The seniors participating in our study gave us similar feedback. However, we also indicated differences between seniors resulting from socio-demographic characteristics. Among other things, an important observation was that active people make active tourists.

Round 2

Reviewer 3 Report

I am now satisfied with the revision and changes introduced in the article.